# Lack of GABARAP-Type Proteins Is Accompanied by Altered Golgi Morphology and Surfaceome Composition

**DOI:** 10.3390/ijms22010085

**Published:** 2020-12-23

**Authors:** Julia L. Sanwald, Jochen Dobner, Indra M. Simons, Gereon Poschmann, Kai Stühler, Alina Üffing, Silke Hoffmann, Dieter Willbold

**Affiliations:** 1Institut für Physikalische Biologie, Heinrich-Heine-Universität Düsseldorf, Universitätsstraße 1, 40225 Düsseldorf, Germany; julia.sanwald@uni-duesseldorf.de (J.L.S.); j.dobner@fz-juelich.de (J.D.); indra.simons@uni-duesseldorf.de (I.M.S.); a.ueffing@fz-juelich.de (A.Ü.); 2Institute of Biological Information Processing (IBI-7: Structural Biochemistry), Forschungszentrum Jülich, Leo-Brandt-Straße, 52428 Jülich, Germany; 3Institute of Molecular Medicine I, Proteome Research, Heinrich-Heine-Universität Düsseldorf, Universitätsstraße 1, 40225 Düsseldorf, Germany; gereon.poschmann@uni-duesseldorf.de (G.P.); kai.stuehler@uni-duesseldorf.de (K.S.); 4Molecular Proteomics Laboratory, Biologisch-Medizinisches Forschungszentrum (BMFZ), Heinrich-Heine-Universität Düsseldorf, Universitätsstraße 1, 40225 Düsseldorf, Germany

**Keywords:** Atg8, GABARAP, Golgi apparatus, surfaceome

## Abstract

GABARAP (γ-aminobutyric acid type A receptor-associated protein) and its paralogues GABARAPL1 and GABARAPL2 comprise a subfamily of autophagy-related Atg8 proteins. They are studied extensively regarding their roles during autophagy. Originally, however, especially GABARAPL2 was discovered to be involved in intra-Golgi transport and homotypic fusion of post-mitotic Golgi fragments. Recently, a broader function of mammalian Atg8s on membrane trafficking through interaction with various soluble *N*-ethylmaleimide-sensitive factor-attachment protein receptors (SNAREs) was suggested. By immunostaining and microscopic analysis of the Golgi network, we demonstrate the importance of the presence of individual GABARAP-type proteins on Golgi morphology. Furthermore, triple knockout (TKO) cells lacking the whole GABARAP subfamily showed impaired Golgi-dependent vesicular trafficking as assessed by imaging of fluorescently labelled ceramide. With the Golgi apparatus being central within the secretory pathway, we sought to investigate the role of the GABARAP-type proteins for cell surface protein trafficking. By analysing the surfaceome composition of TKOs, we identified a subset of cell surface proteins with altered plasma membrane localisation. Taken together, we provide novel insights into an underrated aspect of autophagy-independent functions of the GABARAP subfamily and recommend considering the potential impact of GABARAP subfamily proteins on a plethora of processes during experimental analysis of GABARAP-deficient cells not only in the autophagic context.

## 1. Introduction

The autophagy-related 8 (Atg8) proteins, consisting of members of the microtubule-associated proteins 1A/1B light chain 3 (MAP1LC3, hereafter LC3) subfamily and the γ-aminobutyric acid type A (GABA_A_) receptor-associated protein (GABARAP) subfamily, are mainly recognised for their functions during autophagy.

In that context, the GABARAP-type proteins were shown to be involved in the later steps, involving autophagosome closure [1] and autophagosome-lysosome fusion [2], thereby enabling the autophagic degradation and recycling of cellular components. Interdependence of autophagy, endocytosis, and secretion pathways has been reported by a growing number of studies (reviewed in [3]). Notably, the three members of the GABARAP family, GABARAP, GABARAP-like 1 (GABARAPL1), and GABARAP-like 2 (GABARAPL2), were initially discovered in the context of transport and trafficking processes. Early studies described an association of GABARAP with the eponymous GABA_A_ receptor [4], and subsequently with other receptors including, e.g., the transferrin receptor (TFRC) [5]. Participation of GABARAP in vesicular transport along microtubules has also been reported [6,7]. Similarly, GABARAPL1 was also found to associate with tubulin [8] and to facilitate, e.g., κ opioid receptor trafficking [9]. The third member of the GABARAP family, GABARAPL2, was identified as an intra-Golgi transport modulator interacting both with the *N*-ethylmaleimide-sensitive factor (NSF) and the Golgi v-SNARE GOS-28 [10,11], and was reported to be required during post-mitotic Golgi reassembly [12]. In both processes, GABARAPL2 is deemed to act as a Golgi-SNARE protector [13]. Consistent with this idea, recent reports show direct interactions of all LC3 and GABARAP proteins with other types of SNARE proteins [14,15].

Furthermore, GABARAP was shown to interact with NSF [16] and 130 kDa *cis*-Golgi matrix protein (GM130), which tethers a certain pool of GABARAP proteins to the Golgi [17]. Other examples for Golgi-associated proteins interacting with GABARAP include PX-RICS, a splicing variant of Rho GTPase-activating protein (RICS) containing a phosphoinositide-binding (PX) domain that by interaction with GABARAP mediates ER-to-Golgi transport [18], and optineurin, that interacts with various human Atg8 paralogues and is involved in various cellular processes, including Golgi maintenance [19,20]

In non-mitotic mammalian cells, the Golgi apparatus consists of interconnected stacks of cisternae [21,22]. During conventional protein transport and secretion, the Golgi serves as a central trafficking organelle. After translation in the endoplasmic reticulum (ER), respective proteins are loaded into coatomer protein complex-II (COPII)-coated vesicles, transported to an ER-Golgi intermediate compartment (ERGIC), and translocated in an anterograde manner by passing the *cis*-, the *medial*-, and the *trans*-Golgi. Once a protein has reached the *trans*-Golgi network (TGN), it is sorted by coat proteins for its destination, for example the plasma membrane (PM) [23]. However, not only the transport of proteins, but also that of lipids is a central Golgi function. One of these lipids is ceramide, which is transported from the ER to the Golgi by ceramide transfer protein (CERT) [24]. Once it has reached the Golgi, ceramide is metabolised by sphingomyelin synthases. Ceramide metabolites are further transported to the PM or other membranes [25].

In this work, we employed various knockout (KO) cell lines to study the role and importance of the GABARAP subfamily in maintaining Golgi morphology and on lipid transport in a ceramide chase experiment. Finally, we comparatively analysed the surfaceomes of wild-type (WT) cells and of cells deficient for the GABARAP subfamily. Taken together, we demonstrate that the GABARAP subfamily, additionally to its well-described roles during autophagy, is involved in Golgi apparatus morphology maintenance, secretory vesicular trafficking of lipids and cell surface proteins. We thus suggest considering these autophagy-independent effects when analysing GABARAP-type protein function.

## 2. Results

### 2.1. GABARAP- and/or GABARAPL2-Deficient Cells Display Altered Golgi Morphology

To investigate the impact of individual GABARAP family members on Golgi morphology, we used a panel of human embryonic kidney 293 (HEK293) KO cell lines which exhibit a single KO (SKO), a double KO (DKO) combination, or a triple KO (TKO) of the respective GABARAP-type protein gene locus [26,27]. Golgi morphology was studied via visualisation of both the TGN by anti-TGN46 staining (Figure 1A) and the complete Golgi by BODIPY-FL C5-ceramide staining (Appendix A). For both markers, consistent patterns for each of the analysed cell lines were observed, cross validating the observed results. Next, because of its more distinct staining profile, we categorised the signal obtained for TGN46 as compact (I), partly compact (II), and dispersed (III) Golgi pattern (Figure 1B), technically always considering all individual planes of each recorded z-stack during analysis. As summarised in Figure 1C, the vast majority of the stains from WT cells, expressing all three GABARAP-type proteins, were classified as category I or II. Hence, WT cells had a compact or partly compact Golgi morphology in most of the cases (46% and 39%, respectively), while category III patterns indicating extensive Golgi fragmentation were rare. Cells with a GABARAPL1^SKO^ displayed only mild alterations with a slight tendency towards lower Golgi compactness, but overall showed the most WT-like phenotype of all genotypes analysed. In contrast, both GABARAPSKO and GABARAPL2^SKO^ cells showed category I patterns in less than 20% of the cases, and more than 60% the fraction of category II was considerably increased compared to WT. However, the percentage of category III was similar between all the three SKO lines and thus also resembled the WT situation. While SKO-like results were obtained for GABARAPL1/L2^DKO^ cells, GABARAP/L1^DKO^ cells showed a further reduction of category I (to 8%), accompanied with an increase of category II (to 71%). Strikingly, 36% of the GABARAP/L2^DKO^ cells exhibited even more category III Golgi structures than GABARAP/L1/L2^TKO^ cells (33%) and thus displayed the highest degree of disorganisation among all genotypes (Figure 1C). Overall, there was a significant association between the genotype analysed and Golgi compactness as calculated by Pearson’s chi-squared test (χ^2^ (14) = 414.62, *p* < 0.001). Based on the obtained standardised residuals (Figure 1D), which represent the standard deviation of the actual from the expected count, lack of GABARAP or GABARAPL2 alone or in combination with a lack of GABARAPL1 was associated with a shift from a compact to a more dispersed Golgi morphology. Strikingly, GABARAP/L1^DKO^ cells specifically showed an enrichment of partly compact Golgi morphology, whereas GABARAP/L2^DKO^ and GABARAP/L1/L2^TKO^ cells showed an enrichment of dispersed Golgi morphology. The representative 3D visualisations of the recorded confocal stacks after TGN46-staining can be found in Appendix A. Consistently, also staining of the *cis*-Golgi marker protein GM130 (Appendix A), followed by the categorization as applied for TGN46 staining (Figure 1), revealed a shift from compact to partly compact and completely dispersed Golgi morphology in GABARAP/L2^DKO^ and GABARAP/L1/L2^TKO^ cells compared to WT (Appendix A). The same was true for another marker of *cis*-Golgi, Golgi reassembly-stacking protein of 65 kDa (GRASP65), which was analysed accordingly (Appendix A). Taken together, these results indicate that besides GABARAPL2, at least also GABARAP seems to be involved in Golgi maintenance.

### 2.2. GABARAP-Type Protein Deficiency is Accompanied by Impaired Ceramide Trafficking

As the Golgi plays a fundamental role in conventional protein secretion and PM-directed transport [28], and because of the described transport-related functions of GABARAP and its two paralogues [4,9,10], we hypothesised that a lack of all GABARAP subfamily members at once should considerably impact membrane trafficking from the Golgi to the PM. It is well known that ceramide, after being transported by CERT from the ER to the Golgi apparatus, is converted to sphingomyelin, glucosylceramide, and more complex glycosphingolipids before it reaches the PM, and can therefore be used to study lipid metabolism and vesicle-mediated lipid transport from the Golgi to the PM [29]. Fluorescently labelled probes such as 6-((*N*-(7-Nitrobenz-2-Oxa-1,3-Diazol-4-yl)amino)hexanoyl)Sphingosine (NBD C_6_-ceramide) or BODIPY-FL C_5_-ceramide are selective stains for the Golgi in living and fixed cells [30] and therefore are applicable to study both Golgi morphology and Golgi-related lipid transport.

Thus, as schematically depicted in Figure 2A, we followed the subcellular distribution of NBD C_6_-ceramide over time in WT and GABARAP/L1/L2^TKO^ cells in the presence or absence of inhibitors of Golgi-related vesicular trafficking by live-cell fluorescence microscopy. The respective results are summarised in Figure 2B. While after 30 min a rather diffuse labelling of intracellular membranes was observed, after 90 min we noted an increase in staining intensity in the perinuclear region of WT cells, likely representing the Golgi apparatus. In GABARAP/L1/L2^TKO^ cells, on the other hand, the staining appeared in more punctate or vesicular structures compared to WT cells, a pattern which became even clearer after 24 h. Although WT cells also showed vesicular staining, they additionally exhibited staining at the PM which was only very faint in GABARAP/L1/L2^TKO^ cells. After 48 h, overall staining became less intense. At this timepoint, WT cells exhibited faint PM staining and diffuse intracellular labelling, while in GABARAP/L1/L2^TKO^ cells, still intensely labelled vesicular structures were found. When incubating the cells for 90 min with 10 µM Brefeldin A (BFA), ceramide additionally labelled perinuclear rims probably representing ER staining due to ER-to-Golgi fusion by BFA [31]. Interestingly, the overall staining pattern appeared similar between WT and GABARAP/L1/L2^TKO^ cells in the presence of BFA. Since BFA only inhibits vesicular transport without affecting ER-to-Golgi lipid transport by CERT [31], this indirectly suggests that GABARAP subfamily proteins do not have an impact on non-vesicular ER-to-Golgi transport of ceramide.

Finally, incubation for 24 h with 10 µM Monensin, an ionophore known to cause disruption of the *trans*-Golgi apparatus [32], led to a similar phenotype in WT and GABARAP/L1/L2^TKO^ cells, resembling the ceramide distribution in GABARAP/L1/L2^TKO^ cells after 24 h without Monensin. Although, in principle, Monensin treatment can affect cell viability by causing oxidative stress [33], after 24 h of staining, cell nuclei were still intact, indicating that the cells were still viable. Taken together, these results hint towards a role of the GABARAP subfamily at least in *trans*-Golgi-to-PM trafficking of ceramide and its metabolites.

### 2.3. GABARAP-Type Protein Deficiency is Associated with Altered Surfaceome Composition

To investigate whether compromised Golgi integrity and impaired anterograde ceramide transport in the absence of the GABARAP subfamily is accompanied by altered cell surface protein expression, we finally analysed the surfaceome of GABARAP/L1/L2^TKO^ cells in comparison to WT cells.

To compare their cell surface proteomes, cultures of WT and GABARAP/L1/L2^TKO^ cells were exposed to surface biotinylation, lysed, subjected to streptavidin pull-down, and the protein pools obtained were processed by quantitative proteomics. Data were post-processed (for details refer to material and methods section) and normalised protein levels were analysed. As shown in Figure 3A (for raw data please refer to Appendix A), a total of 2710 different proteins were identified by this approach. By concentrating on already verified surface-located proteins, 216 of the initially identified hits were found to be surface annotated with high confidence (verified or putative) according to the cell surface protein atlas (CSPA, [34]). Notably, with 58 proteins, more than 25% of them displayed significantly different abundances and were therefore examined in more detail. Remarkably, 36 proteins displayed a significantly higher and 22 proteins displayed a significantly reduced surface abundance in GABARAP/L1/L2^TKO^ compared to WT cells. Figure 3B shows hierarchical clustering of these proteins, thereby demonstrating the high degree of conformity between individual replicates and the identity of the respective associated proteins (for more detailed information please refer to Appendix A).

Among all differentially abundant surface annotated proteins, a gene ontology (GO)/reactome pathway enrichment analysis revealed, among others, significant over-representation of proteins belonging to the category of sodium ion membrane transport (−log10 *p* = 3.18, 37.8-fold,) and cell adhesion (−log10 *p* = 1.41, 6.7-fold) as given in Figure 3C,D, respectively.

Most of the identified annotated surface proteins were single pass type I transmembrane proteins (32/58), but also single pass type II transmembrane proteins (4/58), multi pass transmembrane proteins (13/58), and GPI-anchored membrane proteins (2/58) were identified. Proteins with significantly higher abundance in GABARAP/L1/L2^TKO^ compared to WT cells included those with described or predicted transporter/channel activity or associated proteins thereof (CNNM4, CNNM2, SLC4A7, ASIC1, ABCC1, ANO6, SLC39A14, ATP1B3, ATP1A1, SLC3A2, ITPRIP) as well as receptor or receptor-associated proteins (ITGA7, ADAM15, INSR, PTPRF, TYRO3, M6PR, TFRC), cell adhesion associated proteins (PODXL2, KIRREL, NPTN, ALCAM/CD166, JAM3, CADM1), or proteins with described involvement in immunity (CD59, ALCAM/CD166, NCR3LG1, CD276, CADM1).

Proteins with known ER and/or Golgi association (GGCX, TMED7, SMPDL3B, SEL1L) were also identified within the group of higher surface abundance in GABARAP/L1/L2^TKO^ cells. Proteins with significantly lower abundance at the PM of GABARAP/L1/L2^TKO^ compared to WT cells were functionally more dispersed and included proteins associated with immunity (HLA-A, HLA-C, CD46), autophagy (CAPNS1), Ca^2+^ channel activity (CACHD1), receptor proteins (EPHA4), adhesion (MCAM/CD146, EPCAM, F11R, LGALS3BP), and proteins associated with ER and/or Golgi (RPN1, MIA3, EMC1, TMEM259, STT3A).

Finally, Figure 3E displays a scatter plot of the 2710 protein hits detected in this study. Among those proteins with significantly increased abundance in cells lacking the GABARAP-type proteins (red dots), we identified with TFRC one of the earliest described GABARAP interactors [5]. Although the functional relevance of this association has not been clarified to date, it is tempting to speculate that its surface trafficking and/or correct glycosylation are dependent on any or all GABARAP-type proteins. Importantly, abundance of the major histocompatibility I (MHC-I) subtypes HLA-A and HLA-C was found to be reduced in GABARAP/L1/L2^TKO^ cells. This is intriguing, because macroautophagy has been implicated in MHC-I antigen presentation to the cell surface [35]. As a proof of concept, we monitored TFRC and HLA-A surface levels by IB of biotinylated and pulled surface-enriched proteins and fluorescence-activated cell sorting (FACS), respectively. Surface levels of TFRC were significantly increased (Appendix A), while surface HLA-A levels were significantly decreased (Appendix A) in GABARAP/L1/L2^TKO^ compared to WT cells, which is in agreement with the MS data (Appendix A and related Figure 3B,E).

A further interesting hit is the acid sphingomyelinase-like phosphodiesterase 3b (SMPDL3B) which was described to regulate the levels of ceramide metabolites ceramide-1-phosphate (C1P) and sphingosine-1-phosphate (S1P) [36]. SMPDL3B is a lipid raft associated enzyme and thus involved in regulation of membrane fluidity [37]. Notably, another member of the family of sphingomyelinases, the neutral sphingomyelinase 2 (nSMase2), has been linked to LC3, which directly interacts with the nSMase2 regulator FAN (factor associated with nSMase2 activation) to specify cargo loading, e.g., of heterogeneous nuclear ribonucleoprotein K (HNRNPK), into extracellular vesicles (EVs) [38]. Very recently, we detected HNRNPK also in GABARAP-containing EVs [39]. Intriguingly, here we identified increased abundances of both SMPDL3B and HNRNPK in the absence of GABARAP-type proteins. Whether HNRNPK secretion is directly influenced by the GABARAP subfamily will thus be very interesting to determine in the future.

## 3. Discussion

In this work, we demonstrate that the lack of GABARAP, GABARAPL1, and GABARAPL2, both individually and combined (DKO, TKO), alter Golgi morphology, and that a lack of the whole GABARAP subfamily influences important vesicle-mediated intracellular trafficking events like ceramide distribution and surface protein expression. Our results thus extend the current knowledge regarding GABARAP-type protein functions in Golgi-related processes, which was limited to GABARAPL2 so far [13], to the two closely related paralogues GABARAP and GABARAPL1, which have not been investigated in this context yet.

First, we show that in addition to the expected effect of GABARAPL2 deficiency on Golgi morphology, also a lack of GABARAP significantly impacts the Golgi, shifting it from a compact to a more dispersed phenotype. Based on transcriptional data of HEK293 cells [40], it can be assumed that both proteins are expressed at comparable levels, thus suggesting that GABARAP and GABARAPL2 act in parallel to maintain the Golgi structure. This notion is further supported by the fact that double deficiency for GABARAP and GABARAPL2 drastically increased the Golgi disorganisation observed in this study, which would not be explainable if either protein was the only key player in this regard. The minor effect seen for GABARAPL1-deficient cells in this context might be based on the lower abundance of this paralogue in HEK293 cells, which can also be assumed from available gene expression data [40].

Golgi fragmentation and its functional impairment have been reported to be not necessarily causative, as in many Golgi fragmentation phenotypes cell surface transport processes function at normal kinetics [41,42,43]. Furthermore, it has been shown that Golgi fragmentation induced by gold nanoparticles, although not compromising the viability of individual cells, negatively affects cellular adhesion [44]. However, recently, DKO of GRASP55 and GRASP65 was reported to result in Golgi fragmentation accompanied by functional impairment [45]. Increased Golgi fragmentation has also been reported in the context of, e.g., cancer and neurodegenerative diseases such as Parkinson’s disease or amyotrophic lateral sclerosis (ALS) as reviewed in [46]. Notably, also a KO of the GABARAP interactor GM130 led to a disruption of the Golgi, causing trafficking defects in mice [47]. Therefore, it is conceivable that alterations in Golgi morphology caused by a GABARAP-type protein deficiency as shown in this study might also have functional implications.

In line with this notion, by studying fluorescently labelled ceramide as a well-characterised example for Golgi-mediated vesicular trafficking, we discovered impaired PM-directed transport of fluorescently labelled ceramide and its metabolites in cells deficient for all three GABARAP-type proteins.

BFA treatment, which is known to inhibit vesicular ER-to-Golgi trafficking by fusion of ER with Golgi, had no additional effect on TKO cells, indicating that non-vesicular transport of ceramide by CERT seems to be unaffected in the absence of the GABARAP subfamily.

Strikingly, treatment of WT cells with the ionophore Monensin, which is known to cause a disruption of the *trans*-Golgi apparatus [32], mimicked the effect of GABARAP/L1/L2^TKO^ on ceramide trafficking, indicating that GABARAP subfamily proteins play a role during vesicular transport of ceramide and its reaction products. Furthermore, in GABARAP/L1/L2^TKO^ cells we observed a dispersed fluorescence signal when staining not only the TGN marker TGN46, but also with the pan-Golgi marker BODIPY-FL C_5_-ceramide. We thus speculate that further Golgi compartments rely on the presence of at least one GABARAP paralogue to maintain compactness. An obvious explanation for these results is provided by a Golgi-SNARE protector function already suggested for GABARAPL2 [13] which prevents uncontrolled (re-)fusion of Golgi-associated membranes, and which may be redundant for GABARAP and/or GABARAPL1.

Another explanation might be given by the reported interaction of GABARAP with the phosphatidylinositol 4-kinase IIα (PI4KIIα) [48]. PI4KIIα produces the messenger lipid phosphatidylinositol-4-phosphate (PI4P) which is, e.g., implicated in endosomal trafficking [49,50]. Locally increased PI4P levels furthermore lead to an accumulation of Golgi-derived endosomes [51]. GABARAP-type proteins might be involved in subcellular targeting of PI4P by PI4KIIα and thus influence local lipid homeostasis. Absence of GABARAP-type proteins might lead to PI4KIIα mislocalisation which could explain the accumulation of intensely NBD C_6_-ceramide-labelled structures observed in this work.

Consistent with the hypothesis that Golgi fragmentation is linked with disturbed membrane transport, our comparative analysis of the surfaceomes of WT and GABARAP/L1/L2^TKO^ cells revealed a substantial number of proteins with significantly different surface abundance. This suggests that the trafficking of surface proteins is influenced by the GABARAP subfamily in a far more general manner than supposed to date. The fact that some of the surface-located proteins identified during our proteomics study were up-regulated, while others were downregulated, is particularly interesting and most likely reflects the plethora of cellular processes that GABARAP-type proteins participate in, mainly as providing interaction platforms for protein complexes [52,53,54,55]. It must thus be considered that GABARAP-type protein deficiency provokes a pleiotropy of potentially counteracting effects. For example, a reduction of overall degradation kinetics due to the function of the GABARAP-type proteins during lysosomal fusion events [2,56] might be counteracted by non-redundant roles of single paralogues, as, e.g., shown by the enhanced degradation of EGFR in the absence of GABARAP [26]. This adds further complexity to the picture and illustrates how several opposing effects might be evoked simultaneously. Accordingly, altered Golgi dynamics, probably in concert with inputs of further processes, likely determine the actual degree of the surface abundance of an individual protein. It is also conceivable that the diverse processes vary in their impact depending on the nature of each surface protein affected and the respective metabolic status of the cell system investigated. In parallel, deficiency of the GABARAP-type subfamily may additionally directly disturb the intracellular distribution of surface proteins, especially those containing functional interaction motifs [52]. Interestingly, recent results from a yet unpublished study show altered surfaceome composition of *Atg5^-/-^* mouse embryonic fibroblasts [57]. ATG5 is a key component of the LC3/GABARAP lipidation machinery which is essential for their integration into autophagy-related and unrelated membranes [53]. The effect of GABARAP subfamily protein lipidation on cell surface protein trafficking will be interesting to determine in future studies.

However, it has to be kept in mind that many surface proteins have been described to bypass the canonical Golgi secretion pathway [28]. Particularly, autophagy-dependent secretion [58] is likely influenced by a lack of GABARAP subfamily proteins. Deficiency of GABARAP-type proteins might furthermore result in Golgi bypass of proteins which are usually secreted conventionally. This would lead to a subset of dysfunctional PM-associated proteins due to altered glycosylation patterns [59].

In summary, the presented work demonstrates a significant impact of GABARAP subfamily proteins on Golgi morphology, ceramide trafficking and surfaceome composition in cultured cells. The variety of proteins in terms of molecular function with altered surface abundance is broad and illustrates how many processes are potentially affected by the absence of all GABARAP-type proteins. We therefore conclude that the lack of the GABARAP subfamily is associated with impairment of a multitude of processes on a cellular and likely also organismal level. Hence, we suggest consideration of general cellular integrity which might be compromised on several levels. Attention should not be limited to autophagy, but also be given to phenotypical Golgi morphology and, if applicable, also to the extent of surface expression and functionality of selected proteins when working with GABARAP subfamily-deficient systems.

## 4. Materials and Methods

### 4.1. Antibodies

For immunofluorescence, primary TGN46 antibody (Cat. No. AHP500GT, Bio-Rad Laboratories, Hercules, CA, USA) was used at a concentration of 1:250. Goat anti-GRASP65 antibody (Cat. No. sc-19481, Santa Cruz Biotechnology, Dallas, TX, USA) and goat anti-GM130 antibody (Cat. No. sc-16268, Santa Cruz Biotechnology, Dallas, TX, USA) were used at a concentration of 1:50. Sheep-488 (Cat. No. A-11015, Thermo Fisher Scientific, Waltham, MA, USA) and Donkey Anti-Goat IgG H&L Alexa Fluor 647 (Cat. No. ab150131, abcam, Cambridge, UK) were used as a secondary antibody at a concentration of 1:200 and 1:250, respectively.

### 4.2. Eukaryotic Plasmids

KO plasmids are based on plasmid pSpCas9(BB)-2A-GFP (PX458) which was a gift from Feng Zhang (Addgene plasmid # 48138) [60].

### 4.3. Cell Culture

Human embryonic kidney 293 Flp-In T-REx (HEK293 Flp-In T-REx; Cat. No. R78007, Thermo Fisher Scientific, Waltham, MA, USA) cells were cultured at 37 °C and 5% CO_2_ in Dulbecco’s Modified Eagle’s Medium (DMEM, Cat. No. D5796, Sigma-Aldrich, St. Louis, MO, USA) supplemented with 10% Foetal Calf Serum (FCS, Cat. No. F9665, Sigma-Aldrich, St. Louis, MO, USA). Cells were routinely checked for mycoplasma contamination.

### 4.4. CRISPR/Cas9 Mediated KO Generation

KO cell lines were generated and validated as described previously [26,27]. In brief, HEK293 Flp-In T-REx cells were transfected with KO plasmids based on pSpCas9(BB)-2A-GFP (PX458) [60] and single sorted for fluorescence protein (FP) positive signals via fluorescence-activated cell-sorting (FACS) in wells of 96 well plates. Clonal lines were recovered, and occurrence of genome editing was verified via amplification of a 400 bp product flanking the target site and sanger sequencing as well as on a protein level with specific antibodies.

### 4.5. Ceramide Chase

The ceramide chase experiment was conducted according to [29]. Briefly, HEK293 Flp-In T-REx cells (3 × 10^5^) were seeded into fibronectin-coated 35 mm imaging dishes (Cat. No. 81158, ibidi, Gräfelfing, Germany) and cultured for 24 h in phenol red-free DMEM (Cat. No. 21063029, Thermo Fisher Scientific) supplemented with 10% FCS. Fluorescent NBD C_6_-Ceramide, Cat. No. N1154, Thermo Fisher Scientific, Waltham, MA, USA) was dissolved in 95% ethanol to a stock concentration of 1 mM. Cells were labelled with 10 nmol/mL NBD C_6_-Ceramide in Hanks’ Balanced Salt Solution (Cat. No. 14025050, Thermo Fisher Scientific, Waltham, MA, USA) containing 0.68 mg/mL bovine serum albumin (BSA) for 1 h at 4 °C in the dark. After labelling, the medium was aspirated, and the cells were rinsed two times with HBSS and further incubated at 37 °C and 5% CO_2_ for 30 min, 90 min, 24 h, and 48 h. As inhibitors of Golgi function, Brefeldin A (Cat. No. 00-4506-51, Life Technologies, Carlsbad, CA, USA) and Monensin (Cat. No. 00-4505-51, Life Technologies, Carlsbad, CA, USA) were used at a concentration of 10 µM each. Cells were incubated with Hoechst 33342 (Cat. No. R37605, Invitrogen, Carlsbad, CA, USA) according to the manufacturer’s instructions for nuclei staining.

### 4.6. Immunofluorescence (IF)

HEK293 Flp-In T-REx cells (3 × 10^5^) were seeded in the presence of 1% penicillin/streptomycin (Cat. No. P0781, Sigma-Aldrich, St. Louis, MO, USA) into fibronectin-coated 35 mm imaging dishes (Cat. No. 81158, ibidi, Gräfelfing, Germany) and cultured for 24 h in DMEM supplemented with 10% FCS. For immunostaining with anti-TGN46 antibody, Flp-In T-REx 293 cells (Cat. No. R78007, Invitrogen, Carlsbad, CA, USA) (WT and as GABARAP(s) SKO, DKO, or TKO) were fixed at 37 °C for 10 min with 4% (*w*/*v*) paraformaldehyde (PFA; pH 6.5), washed two times with PBS, pH 7.4, and permeabilised by shaking in 0.2% Triton-X-100 for 30 min at RT. Cells were blocked by incubation in 1% BSA over night at 4 °C and incubated on the following day first with primary antibody for 1 h shaking at RT, washed three times with PBS, and then incubated with secondary antibody for 1 h shaking at RT under exclusion of light. Again, the cells were washed two times, stored in long storage buffer (0.05% sodium azide in PBS), and applied to image acquisition.

### 4.7. Image Acquisition–Laser Scanning Microscopy (LSM)

For image acquisition, an LSM 710 confocal microscope (Zeiss, Oberkochen Germany) equipped with ZEN black 2009 software (Zeiss, Oberkochen, Germany) and a Plan-Apochromat 63x/1.40 Oil DIC M27 objective was used. Nuclei (DAPI or Hoechst 33342) were visualised using the 405 nm channel (MBS -405), TGN46 using the 488 nm channel (MBS 488) and NBD C_6_-Ceramide using the 458 nm channel (MBS 458). The number of focal planes (z-frames) with a z-distance of 0.4 µm was set between 11 and 18 or 11 and 34 for the recording of TGN46- or BODIPY-stained cells, respectively.

### 4.8. Image Evaluation

Image analysis was done using ImageJ/Fiji [61,62]. All individual planes of the z-stacks recorded were combined in ImageJ by applying the function „SUMSLICES”. Morphology of *trans*-Golgi was qualitatively judged for each cell visually by categorizing the obtained TGN46-staining patterns according to Figure 1B. The 3D visualisations of the recorded confocal stacks from the TGN46-stains were obtained by using ZEN 2.3 SP1 FP1 (black edition). All images have been arranged using CorelDRAW 2017 (version 20, Corel Corporation, Ottawa, Canada). Data analysis and visualisation were done using GraphPad Prism (version 8, GraphPad Software, San Diego, CA, USA). Pearson’s chi-square test statistic and standardised residuals representing z-scores were calculated using the statistical analysis software package (SPSS, version 22, SPSS Inc., Chicago, IL, USA).

### 4.9. Isolation, Identification, Quantification and Analysis of Surfaceomes

PM-based proteins were isolated as described before using the Pierce Cell Surface Protein Isolation Kit according to the manufacturer’s instructions (89881, Thermo Fisher Scientific, Waltham, MA, USA) [26].

Briefly, for each individual experiment, four T75 flasks were prepared, pooled and further processed at >80% confluency. On the day of surface protein isolation, flasks were labelled with Sulfo-NHS-SS Biotin at 4 °C on a shaker. After quenching, cells were pelleted and lysed with periodical sonication and vortexing steps. Biotinylated surface proteins were bound to NeutrAvidin beads, eluted and prepared for mass spectrometric measurements by in-gel digestion essentially as described [63]. Briefly, proteins were separated over a short distance (about 5 mm) in a polyacrylamide gel, stained, reduced with dithiothreitol, alkylated with iodoacetamide and digested with trypsin overnight. Peptides were extracted from the gel and reconstituted in 0.1% (*v*/*v*) trifluoroacetic acid in water. Liquid chromatography coupled with mass spectrometry were essentially carried out as described [63]. Then, 500 ng peptides per sample were separated using a 2 h gradient on C18 material using an Ultimate 3000 Rapid Separation Liquid Chromatography system (Thermo Fisher Scientific, Waltham, MA, USA) online coupled via a nano-source electrospray interface to a QExactive plus (Thermo Fisher Scientific, Waltham, MA, USA) mass spectrometer operated in positive data dependent mode. First, survey scans were recorded at a resolution of 70,000 and subsequently, up to 10 two- and three-fold charged precursors were selected by the quadrupole of the instrument (2 m/z isolation window), fragmented by higher-energy collisional dissociation and analysed at a resolution of 17,500.

Recorded mass spectra were further analysed by MaxQuant (version 1.6.2.10, Max Planck institute for biochemistry, Planegg, Germany) enabling peptide and protein identification and label-free quantification (LFQ). Searches were carried out with standard parameters if not indicated otherwise and were based on 73,112 protein entries from the homo sapiens reference proteome (UP000005640, downloaded on 18 August 2018 from the UniProt Knowledgebase). Label-free quantification was enabled as well as the ‘match between runs’ option. Peptides and proteins were identified at a false discovery rate of 1% and only proteins considered for further analysis showing at least 2 different peptides.

Positive hits were inferred when at least three valid values were detected in at least one group (WT or GABARAP/L1/L2^TKO^). Log_2_ transformed intensities were normalised by subtracting the median from every value. Afterwards, missing values were imputed by replacing them with random values from the normal distribution (downshift 1.8 SD, width 0.3 SD). One n (WT) was removed, because principle component analysis revealed lack of similarity to the other WT samples. Two-tailed two-sample Student’s T-test was calculated (S0: 0, FDR: 5%) for a surface-annotated subset according to the CSPA [34]. Hierarchical clustering was calculated (Euclidean distance, pre-processed with κ-means, average linkage) for differentially expressed annotated surface proteins. High confidence surface proteins (CSPA annotated) with significantly different abundances between WT and GABARAP/L1/L2^TKO^ analysed for relative enrichment of gene ontology (GO)/reactome terms where all identified proteins during mass spectrometry were set as a background database.

## Figures and Tables

**Figure 1 ijms-22-00085-f001:**
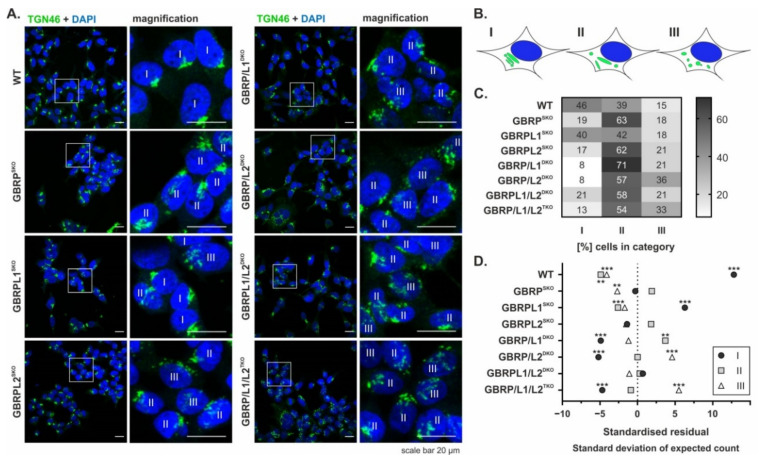
Influence of γ-aminobutyric acid type A (GABA_A_) receptor-associated protein (GABARAP)-type protein deficiency on *trans*-Golgi morphology. (**A**) Wild type (WT) cells or cells deficient for one (SKO), two (DKO), or all three (TKO) GABARAP-type proteins (GBRPs) were fixed (4% PFA), immunolabelled with anti-human TGN46 antibody, and visualised by confocal fluorescence microscopy. Nuclei were counterstained with DAPI. Scale bar, 20 µm. (**B**) Scheme representing the categorisation of a compact (I), partly compact (II), and dispersed (III) Golgi structure. (**C**) Heatmap of percentage of cells per cell type assigned to Golgi category I, II, and III. Per cell type, in total ≥188 cells from ≥5 individual experiments were analysed. Cells were categorised by visual judgement. (**D**) Standardised residual values. Asterisks indicate significant differences from the mean based on the standardised residual distribution with: |z| ≥ 2.58 ** (*p* ≤ 0.01), |z| ≥ 3.29 *** (*p* ≤ 0.001).

**Figure 2 ijms-22-00085-f002:**
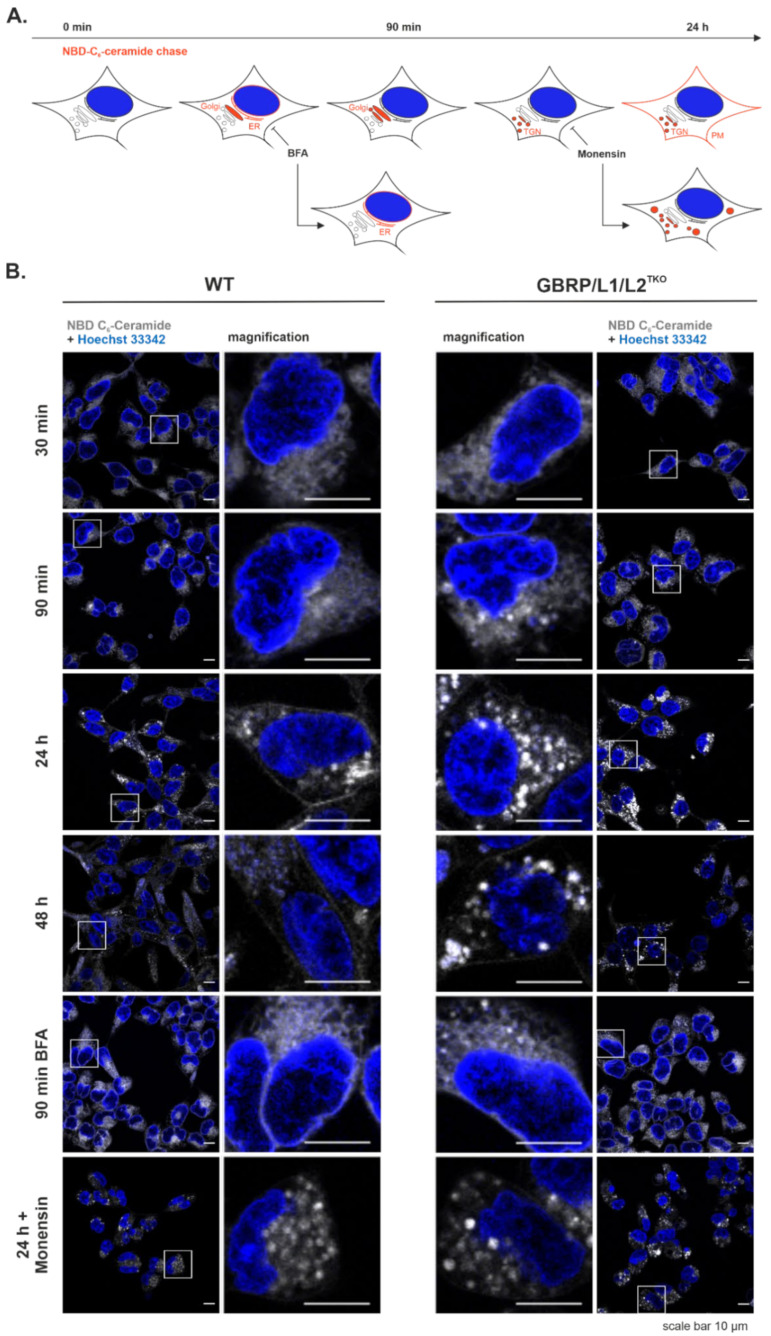
Intracellular ceramide distribution in WT and GABARAP/L1/L2^TKO^ cells. (**A**) Scheme representing the putative intracellular trafficking of NBD C_6_-ceramide (shown here in red) in WT cells, including sites of action of BFA and Monensin. (**B**) Live cell confocal fluorescence microscopy of WT and GABARAP/L1/L2^TKO^ cells labelled with NBD C_6_-ceramide (shown here in grey scale). After labelling for 1 h at 4 °C, the cells were further incubated at 37 °C for 30 min, 90 min, 24 h, or 48 h in full medium, or 90 min in full medium containing 10 µM Brefeldin A (BFA), or 24 h in full medium containing 10 µM Monensin. Nuclei were counterstained with Hoechst 33342. For each condition, one representative image from 5 individual experiments is shown. Scale bar, 10 µm.

**Figure 3 ijms-22-00085-f003:**
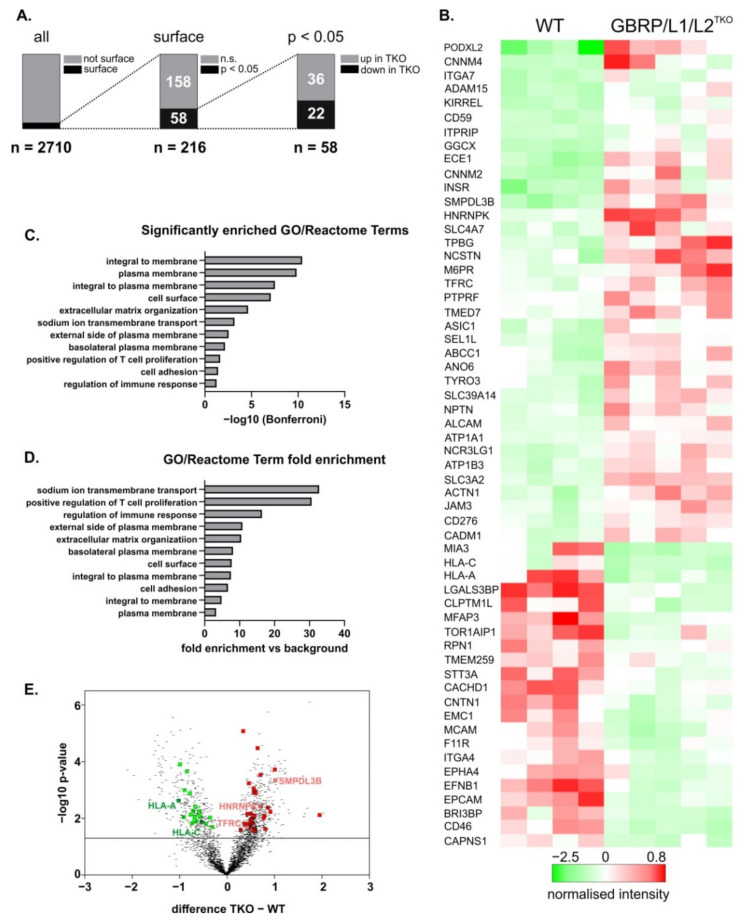
Comparative surfaceome composition analysis between WT and GABARAP/L1/L2^TKO^ cells. (**A**) Representation of filtering and statistical testing workflow. Mass spectrometry of isolated cell surface proteins revealed 2710 proteins of which 216 were surface annotated with high confidence (verified or putative) according to the cell surface protein atlas (CSPA) [34]. Of these 216, 58 showed significantly different abundances between WT and GABARAP/L1/L2^TKO^ cells. (**B**) Heatmap visualising the hierarchical clustering of normalised abundances for the 58 proteins described in (**A**) considering the individual replicates. (**C**,**D**) Relative enrichment of categories of identified proteins as determined by Gene Ontology (GO)/reactome pathway enrichment analysis. (**E**) Scatter plot of independent t-test results of the 2710 proteins. Proteins with significantly higher abundance in GABARAP/L1/L2^TKO^ cells compared to WT cells are marked in red and proteins with significantly lower abundance are marked in green. Proteins further addressed are highlighted. In general, proteins are denoted by their gene names.

## Data Availability

The data presented in this study are available in the article and Appendix A.

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
