# Peer review of "Lack of GABARAP-Type Proteins Is Accompanied by Altered Golgi Morphology and Surfaceome Composition"

_ijms, 2020, doi:10.3390/ijms22010085_

Round 1

Reviewer 1 Report

In their study, the authors evaluate the functions of the GABARAP family members in maintaining Golgi structure, ceramide transport and metabolism as well as protein transport to plasma membrane. They show that GABARAPL2 and/or GABARAP play key functions in these processes uncovering potential new functions beyond their role in autophagy. The paper is well written, the findings well presented and convincing.

The reviewer has comments, suggestions and questions to the authors and hopes that this will help to further improve the quality of the manuscript.

1) Remove lines 89 to 91 (remaining from templates or instructions to authors)

2) Although the authors refers to their recent publication (Dobner et al, Cells, 2020), the reviewer suggests to add in supplementary figure 1 data showing GABARAP family protein expression in the cell lines that are used, potentially using Bafilomycin to limit autophagic degradation and improve detection. 

Has the authors performed any rescue experiment by re-expressing GABARAP, L1 or L2?

3) The authors argue that GABARAPL1 KO has only a minor impact due to low GABARAPL1 expression in HEK cells. Performing the experiment described in comment 2) would support the author's statement. 

4) Several figure references issues can be found in the manuscript: lines 150, 169, 189, 195, 202, 2020.

5) Monensin treatment can be harsh on cells. Was cell viability/adherence of cells affected?

6) The proteomic analysis of plasma membrane composition provides with interesting hits and raises several questions:

  • why did the authors chose to perform this analysis in the TKO cell line and not in the GABARAP/L2 DKO cells which shows similarly altered Golgi?

  • Even if studying the mechanisms explaining why/how the PM localization of some protein is modified in the absence of GABARAPs is beyond the scope of the paper, the reviewer suggests to confirm the altered localization of the 3 proteins that the authors further address in the manuscript (TFRC, SMPDL3B and HNRNPK), either by immunofluorescence or by sub cellular fractionation and WB.
  • This later point could be performed in the TKO and GABARAP/L2 DKO reinforce the findings that GABARAP/L2  are most likely to play a critical role in the process.

Reviewer 2 Report

In this work, the authors address the effect of GABARAP- type proteins on Golgi morphology and on the levels of surface proteins. For that, the authors use knockout (KO) cells for the different GABARAPs, single, double or triple KO, analyse TGN46 and Ceramide positive structures (by BODIPY-FL C5-ceramide and NBD C6-Ceramide) and performed MS in surface biotinylated samples. Despite the interesting data, the study lacks some results/ controls and its merely observational do not presenting any data that explains the molecular mechanisms behind the observations. Considering this, the novelty of this study is limited as is, and some of the conclusions are not totally supported by the data.

Despite the understandable use of the KO approaches, the lack of a protein (SKO) or entire group of proteins (TKO) can have pleiotropic effects, specially at long term. With that, it is legitimate to assume, that the consequences observed upon GABARAP absence, namely Golgi morphology and protein expression at cell surface, results from an indirect effect of this manipulation and not from the role of GABARAP itself. To overcome this, the results should be confirmed upon transient and partial depletion of GABARAP by approaches like siRNA or shRNA. Moreover, the hypothesis/theory that these observations result from an indirect effect, should be patented in the discussion or additional results should be provided to ascertain a direct role of GABARAP on Golgi morphology and surfaceome composition. Interestingly, cell surface-located proteins expression is differential, with some presenting higher and others lower expression than the wildtype cells, a topic that, in this reviewer opinion, should be matter of discussion, giving the authors space to speculate what are the mechanism that  justify these results.

General comments:

  • Characterization of Golgi morphology should be completed with labelling of additional Golgi markers (different Golgi subsets) e.g. cis-Golgi marker GM130, β-COP, 58K, GalT, Giantin. Moreover, categorization of the TGN should be performed taking into account the whole structure (3D) and not a single plane ( this should be clear: “Cells stained with anti-TGN46 were categorised by visual judgement of the individual focal planes to assign a certain Golgi morphology” in Materials and Methods vs “technically always considering all individual planes of each recorded z-stack during analysis”). Representative 3D reconstructions of the z-stacks should be shown. Additionally, techniques more quantitative like Pulse shape analysis (PulSA) (Toh WH, et al. 2015; PMID: 25702121) or which provide better resolution (super-resolution three-dimensional structured illumination microscopy (3D-SIM) or Three-dimensional scanning transmission electron microscopy tomography (STEM-T)) should be considered.
  • Considering the surfaceome composition study, authors should validate the MS results, by analysing plasma membrane levels (by surface biotinylation followed by WB) and/or subcellular distribution of, at least, a significantly higher and a significantly reduced surface protein.
  • Taking into account the importance of Golgi function/structure to autophagy, and central role of GABARAPs in this process, authors should evaluate the status of this degradation process on these cells (autophagic flux), as a proof of concept.
  • It would also be of interest, to study the status (levels and/or distribution) of proteins already demonstrated to be crucial or to modulate Golgi structure (e.g. golgins (like GCC88 and GCC185), GRASPs (like GRASP55), cis-Golgi matrix protein GM130, EHD3, among others) or look for possible targets in the MS data, in an attempt to connect the lost of GABARAPs with the phenotype observed.

Minor comments:

  • the guide RNAs (gRNA) used for KO generation should be indicated and data concerning KO verification should be shown.
  • protocol used for BODIPY-FL C5-ceramide staining is missing
  • visualization/perception of fluorescence images could be improved by the display of grey levels (exclude nucleus) instead of pseudocolored and the use of a different colour look-up table (LUTs) like a rainbow LUT.

Round 2

Reviewer 2 Report

I acknowledge and accept the corrections and additional data presented by the authors that increased the quality of the manuscript.